# Stimulated Thermal Scattering in Two-Photon Absorbing Nanocolloids under Laser Radiation of Nanosecond-to-Picosecond Pulse Widths

**DOI:** 10.3390/nano12152567

**Published:** 2022-07-26

**Authors:** Alexander I. Erokhin, Nikolay A. Bulychev, Egor V. Parkevich, Mikhail A. Medvedev, Igor V. Smetanin

**Affiliations:** 1P. N. Lebedev Physical Institute of RAS, Leninskii prosp. 53, 119991 Moscow, Russia; aerokhin@sci.lebedev.ru (A.I.E.); parkevichev@lebedev.ru (E.V.P.); medvedevma@lebedev.ru (M.A.M.); 2Moscow Aviation Institute, Volokolamskoe shosse 4, 125993 Moscow, Russia; nbulychev@mail.ru

**Keywords:** nonlinear optics in nanoparticle’s colloid, stimulated Rayleigh–Mie scattering, phase conjugation, four-wave mixing, synthesis of nanoparticle in plasma discharge

## Abstract

Recent discoveries in nonlinear optical properties of nanoparticle colloids make actual the challenge to lower the energy threshold of phase conjugation and move it into the domain of shorter pulse widths. A novel effect of the stimulated Rayleigh-Mie scattering (SRMS) in two-photon absorbing nanocolloids is considered as a promising answer to this challenge. We report the results of experimental and theoretical study of the two-photon-assisted SRMS in Ag and ZnO nanocolloids in the nanosecond-to-picosecond pulse width domain. For 12 ns 0.527 μm laser pulses, the four-wave mixing SRMS scheme provides lasing and amplification of backscattered anti-Stokes signal in Ag nanocolloids in toluene at the threshold 0.2 mJ and the spectral shifts up to 150 MHz. For 100 ps 0.532 μm pulses, we observed for the first time efficient (over 50% in signal-to-pump ratio of pulse energies) SRMS backscattering of the anti-Stokes signal in Ag nanocolloids in toluene and predominantly Stokes signal in ZnO nanocolloids in water, with the spectral shifts up to 0.25 cm^−1^. We develop the first order-in-perturbation model of the four-wave mixing two-photon absorption-assisted SRMS process which shows that at nanosecond pulses, amplification is predominantly due to the thermal-induced coherent oscillations of polarization while the slow temperature wave acts also as a dynamic spatial grating which provides a self-induced optical cavity inside the interaction region. At a picosecond pulse width, according to our model, the spectral overlap between pump and signal pulses results in formation of only the dynamic spatial temperature grating, and we succeeded at recovering the linear growth of the spectral shift with the pump power near the threshold.

## 1. Introduction

It has been recognized in past decades that nanoparticle suspensions consisting of various materials, including metal, semiconductor, dielectric, and even organic and biological structures reveal extraordinary nonlinear optical properties [1,2,3]. Observation of Stimulated Rayleigh–Mie Scattering (SRMS) in two-photon absorbing nanocolloids becomes one of the very fascinating novelties in modern nonlinear optics [4,5]. An interest in exploring the possibilities of this new type of nonlinear scattering naturally emerges in view of enhancement of efficiency of phase conjugation.

Originally, the effect of Phase Conjugation (PC) has been discovered on the basis of Stimulated Brillouin Scattering (SBS) when a laser pulse focused into the nonlinear medium is reflected backwards due to coherent excitation of hyper-sound oscillation of medium density [6,7]. Conventionally, to achieve effective PC for laser pulses in the visible frequency range, the pump pulse power and width should exceed ~0.1 MW and ~1 ns, respectively, while the laser spectrum should be narrower than ~1 GHz [8,9]. Besides SBS, most of other known types of stimulated backscattering (i.e., Rayleigh, Raman, thermal, etc.) also allow one to generate phase conjugated waves, but this commonly requires far more rigid pump laser pulse characteristics [10,11].

Observation of Stimulated Rayleigh–Mie Scattering in two-photon absorption media (2PA SRMS) opens a prospective approach to overcome the above restrictions on laser pulse parameters intrinsic to SBS: Using metallic nanoparticle suspensions, it becomes possible to reduce the stimulated scattering threshold power by an order of magnitude [5,12]. These suspensions look prospective in view of lifting the second restriction and allow one to achieve efficient PC in the picosecond time domain.

In our previous reports [13,14], we have reconsidered the experimental results on measured zero spectral shift of 2PA SRMS in nanocolloids which has been reported in [5,12]. Our measurements have been undertaken in Ag nanoparticle colloidal solutions in toluene and hexane and reveal the appearance of non-zero anti-Stokes spectral shifts in backscattered signal for both these colloids. These anti-Stokes spectral shifts were measured up to ~150 MHz being thus appreciably exceeding the Rayleigh line widths. This our finding is in accordance with the first principles of quantum mechanics and statistical physics, which dictate for the gain of a stimulated scattering process to be proportional to the product of the relevant spontaneous differential cross section by the Planck factor $\left(1-\exp(-\hbar \Omega /k_{B}T)\right)$ [7,8]. Therefore, even at small shifts compared to the temperature, $\hbar \Omega /k_{B}T<<1$, the gain should be proportional to the spectral shift Ω of the signal wave and vanish when this spectral shift disappears.

Here, we report on our studies on 2PA SRMS in a nanosecond-to-picosecond time domain which have been conducted with 12 ns 0.527 μm laser pulses in a Ag nanoparticle suspension in toluene, and with 100 -ps 0.532 μm laser pulses in both Ag nanoparticle suspension in toluene and ZnO nanoparticle suspension in water. In the nanosecond pulse width domain, we find the 2PA SRMS threshold to be 0.2–0.4 mJ (at 0.003–0.01 M concentration, respectively) which is an order of magnitude less than the SBS threshold. The anti-Stokes spectral shift is rapidly growing near the threshold, saturates with the peak value of ~150 MHz, and then slowly decreases down to ~80 MHz at 10fold excess of the threshold pulse energy. We succeeded at achieving efficient 2PA SRMS with 100 ps 0.532 μm laser radiation which results in generation of the anti-Stokes signal in Ag nanoparticle suspension in toluene and both in dominant Stokes and smaller anti-Stokes signals in the ZnO nanocolloid in water. In the Ag nanocolloids in toluene, we measured ~0.1 mJ as the threshold pulse energy at 0.003 M concentration, while for the ZnO nanocolloid in water (0.001 M concentration) the threshold is found larger by more than an order of magnitude. The scattering efficiency (signal-to-pulse energy ratio) is measured to be very high reaching ~50% in Ag and even above in ZnO nanocolloids. In the Ag nanocolloid, the anti-Stokes spectral shift depends on the pump pulse energy in the way that is analogous to that for the nanosecond laser pulses, but the shifts values are ~50 times larger reaching values up to 0.25 cm^−1^ which are close to the SBS spectral shifts. We provide a comprehensive theoretical analysis of 2PA SRMS in both the nanosecond and picosecond pump pulse width regimes. We treat it as a nonlinear four-wave mixing process. At nanosecond pulse width, the pump and the backward and forward signal waves are spectrally well-separated, and the two-photon absorption process results in two kinds of temperature perturbations (and coupled to them density perturbations), one of which is the spatially homogeneous coherent oscillation at the beat frequency, and the second is the slow temperature wave propagating at a velocity close to the speed of sound. We show that the amplification effect in the nanosecond interaction regime is provided predominantly by coherent thermal oscillations, while the slow temperature wave acts also as a dynamic spatial grating which provides a self-induced optical cavity inside the interaction region. At picosecond pulse width, spectra of the pump and backward signal pulses overlap, coherent polarization oscillation disappears, and only the dynamic temperature grating provides the scattering and amplification effects.

The paper is organized as follows. Section 2 contains a description of methods we used for the production and characterization of Ag and ZnO nano colloids, the technique of measuring the nonlinear two-photon absorption coefficient *β* of Ag nanoparticle colloids in toluene at the picosecond laser pulse width, the experimental setups on 2PA SRMS at nanosecond and picosecond laser pulse widths. Section 3 contains the results of experimental characterization of 2PA SRMS at nanosecond and picosecond laser pulse widths. In Section 4, the physical mechanisms of 2PA SRMS in nanosecond and picosecond time domains are discussed using the coupled density and temperature fluctuation model. Finally, Section 5 presents the conclusions.

## 2. Materials and Methods

### 2.1. Preparation of Stable Ag and ZnO Nanoparticle Suspensions

To produce Ag and ZnO nanoparticle suspensions we used a new prospective method for the synthesis of nanoscale materials based on combined action of high-intensity elastic ultrasonic vibrations and stationary electric discharges in a liquid medium [15,16]. The resulting acoustoplasma discharge in a cavitating liquid medium possesses specific electro-physical and optical characteristics. This type of plasma has several advantages as a technique for synthesizing nanomaterials—a relatively narrow particle size distribution of the synthesized nanopowder, peculiar composition and properties of the yielding nanomaterials, and relatively high productivity.

Ag nanoparticles were fabricated from silver electrodes according to the procedure described in [15,16] and thus acquired unique surface characteristics due to ultrasonic cavitation which prevents them from secondary agglomeration. Thus, the suspension’s stability time was significantly increased. Stabilizing the liquid medium temperature and the plasma discharge parameters (arc current, voltage, frequency and intensity of ultrasound), we succeeded in fabrication of silver nanoparticle suspensions with ≈2 ± 1 nm radii in toluene. These radii values were determined using the correlation spectroscopy (particle size analyzer Photocor Complex produced by Photocor Ltd., Moscow, Russia) and verified using a Carl Zeiss LEO 912 AB OMEGA scanning transmission electron microscope (LEO Elektronenmikroskopie GmbH, Oberkochen, Germany), see Figure 1.

Samples of ZnO nanoparticles were synthesized using the acoustoplasma discharge technique under the action of intensive ultrasonic cavitation as described earlier [17,18]. As electrode material, zinc wire (99.99%) was used. The synthesis of ZnO nanoparticles was carried out in distilled water. After synthesis, a part of the nanoparticle suspension was precipitated in a centrifuge at 6000 rpm for 10 min, then dried in air at 150 C. Another part was examined as a suspension. Figure 1 shows that the ZnO suspension possesses a quite narrow particle size distribution; the size of particles is in the range from 35 to 50 nm with a distribution peak around 40 nm. These data are confirmed by the examination with scanning electron microscopy: the particles have a characteristic rod-like shape with a longitudinal size of about 40 nm and a transverse size of about 20 nm (Figure 1c,d). SEM images of nanoparticles were obtained on a Carl Zeiss Evo 50 device (Carl Zeiss AG, Oberkochen, Germany), with the silicon dioxide plates used as a substrate. Nanoparticles were applied to substrates by centrifuging the suspension.

Further observations of nanoparticles suspensions shows that they acquire high aggregative and sedimentation stability and do not increase in size over time but form composite associates when applied to a substrate. It is also important to emphasize the absence of adhesion of nanoparticles to each other: the contact zones of the particles are insignificant and do not undergo changes over time, which indicates a weak interaction of particles. This fact gives the possibility to obtain stable dispersed systems again by re-dispersion of nanoparticles previously precipitated and isolated from the medium.

### 2.2. Measurements of the Two-Photon Absorption Coefficient of Laser Radiation in Nanoparticle Colloids

In our experiments on 2PA SRMS we used Ag and ZnO nanoparticle colloids dissolved in toluene and water, respectively, which exhibit rather strong two-photon absorption at our wavelengths of interest, 0.527 and 0.532 μm. Strong 2PA in these materials is related to the resonance excitation of *d*-electrons in Ag while direct-gap zinc oxide ZnO has the band gap of 3.1–3.3 eV, depending on various possessing modifications [19]. Determination of 2PA absorption coefficients for these materials is very important for the interpretation of SRMS experimental results.

Previously, the nonlinear 2PA coefficient *β* for ZnO nanoparticles dissolved in water has been measured by the Z-scan facility to give the rescaled values for our experimental conditions (i.e., ~0.001 M concentration, ~40 nm size) ~1 cm/GW [20,21]. In Ag nanoparticle suspension, the coefficient *β* was measured at 160 fs and 10 ns laser pulse durations, giving the rescaled values for our experimental conditions (i.e., the concentration ~0.005 M, ~2 nm particle size) ~10^−4^ cm/GW and $7.2\times 10^{-2}\;\mathrm{cm}/\mathrm{GW}$, respectively [5]. This discrepancy prompted us to clarify this value and measure nonlinear 2PA coefficient of Ag suspension at the 100 ps pulse duration. To do so, we performed the measurements of nonlinear transmission of 100 ps laser pulses through the cell with 0.005 M Ag nanoparticle colloid, i.e., the dependence of transmission through the cell energy of the laser pulse on its energy at the front edge of the cell.

In our experiments, we used the Lotis TII LS-2151 actively mode-locked Nd:YAG laser (LOTIS-TII Co, Minsk, Belarus) with a second harmonic 0.532 μm single pulse generator. The laser pre-lasing stabilization process results in 100±10 ps (FWHM) pulse duration, and pulse energies up to 12 mJ. We set 5 cm long cell in the waist of the laser beam focused with a 190 cm lens, so that the focus length considerably exceeded the cell dimensions. At the input edge the beam had a Gaussian shape with a radius of 10.9 μm. The results of our transmission measurements at the mJ pulse energy range are shown in Figure 2 along with the images of laser beam shapes at the output edges at low and high pulse energies. Note that output beam shape with an increase in pulse energy becomes far from the Gaussian shape due to nonlinearity in light absorption. We performed emulation of this process assuming independent ray tracing of different radial parts of the laser pulse. For each part, absorption of intensity is guided along the ray by linear (with the coefficient *α*) and two-photon *(β*) absorption processes
(1)dIdx = −α I −β I2,
which results in the following relation for intensity attenuation with the distance *l*
(2)$$I(l)=\frac{I_{0}\exp[-\alpha l]}{1-\left(\beta /\alpha \right)I_{0}(\exp[-\alpha l]-1)},$$
where $I_{0}=I(0)$ is the intensity at the cell entrance. Integrating Equation (2) over the initial Gaussian transverse intensity shape of the beam, one can find the output non-Gaussian shape of the laser pulse. It is thus possible to determine the nonlinear 2PA coefficient *β* taking into account significant depletion of the laser pulse in the nonlinearly absorbing medium. In this way, considering the linear absorption coefficient in toluene α=0.61 cm^−1^ we fit the measured transmissions of the 100 ps laser pulse with $\beta =6.2\times 10^{-2}\;\mathrm{cm}/\mathrm{GW}$ which is close to the value measured at the 10 ns laser pulse width in [5].

### 2.3. Nanosecond 2PA SRMS Experimental Scheme

In our experiments on 2PA SRMS in the nanosecond time domain, we used an Nd-glass laser, passively Q-switched by a GSGG Cr^4+^ crystal and operating at TEM_00q_ mode. After amplification and frequency doubling, it provides us with nearly Fourier-transform-limited laser radiation at the wavelength of 0.527 μm, the pulse energy up to 20 mJ, and the 12 ns FWHM pulse width corresponding to 40 MHz spectral width.

Due to the quasi-continuous character of the interaction processes in the nanosecond domain, it is convenient to study 2PA SRMS with the use of a four-wave mixing scheme rather than the scheme of direct PC generation from noise. To do it, one should split the initial laser pulse into two counter propagating pump waves, one of which has an intensity two or three orders lower than the other but a larger propagation angle spectrum. This allows one to get into the resonance with a couple of counter-propagating scattering signals.

The scheme of our experimental setup is shown in Figure 3. The laser radiation is split by the glass wedge W into two parts of larger and smaller energies, which are then used as the forward (larger energy) and the backward (smaller energy) pump pulses in the four-wave mixing scheme. The forward pump pulse is focused into the 5 cm cell (C) containing the Ag nanoparticle suspension in toluene by the lens F_1_ providing the caustic length of ~1 cm in air. The 2PA SRMS emerges through four-wave mixing between two (forward and backward) pump waves and two (forward and backward) coupled signal waves.

Temporal shapes of the pump and the backward phase-conjugated signal were detected using two photodiodes PD of ~1 ns resolution, and their spectra were also analyzed using the Fabry–Perot interferometer. The latter has the 64 mm base which allows us to distinguish Brillouin and Rayleigh–Mie fractions in the scattered signal. Semi-circles of the pump and the backscattered signal interferometric patterns were simultaneously projected onto a CCD. The digital image was processed with the use of the mathematical procedure described in [22], which allows us to extract the spectral information and resolve spectral shifts as small as ~20 MHz.

### 2.4. Picosecond 2PA SRMS Experimental Scheme

Experimental methods which we utilize in picosecond time domain are similar to those used in nanosecond pulse experiments, but the expected spectral shifts in the picosecond domain are larger by almost two orders of magnitude. Our experimental scheme for the picosecond-SRMS is shown in Figure 4.

The key to measure the 2PA SRMS spectrum characteristics for nanosecond pulses is, along with the proper choice of nanoparticle suspensions, the use of almost Fourier-limited laser radiation. In our picosecond experiments, we succeeded in obtaining similar characteristics. We used the second harmonic of the Lotis LS-2151 Nd:YAG laser with an FWHM pulse width in the range 90–100 ps. The spectrum of this laser was determined using the comprehensive cavity Q-control of the master oscillator. The system was Q-switched and then selected a single optical ultrashort pulse from the master oscillator cavity with its subsequent amplification and frequency conversion into the second harmonic. As a result, we obtained laser pulses at the wavelength λ = 0.532 μm, which have energy up to 10 mJ, 80–100 ps pulse width, and the spectrum width $\Delta \nu_{FWHM}=0.2{\;\mathrm{cm}}^{-1}$.

## 3. Results

### 3.1. Experimental Results of 2PA SRMS in the Nanosecond Time Domain

To detect the SRMS signal in the Ag nanoparticle suspension, we first detected the SBS signal in pure toluene for which we found the threshold was ~2.5 mJ in pump pulse energy. Then, we substituted the pure solvent using the suspensions of Ag nanoparticles with a gradually increasing concentration and found the appearance of an SRMS signal along with the SBS signal, see Figure 5. First of all, we found that the backscattered SRMS signal revealed the anti-Stokes spectral shift. By gradual decrease in the pump pulse energy, we reached the threshold. We found that the SRMS threshold is almost an order of magnitude less than the SBS threshold, varying from 0.4 to 0.2 mJ with an increase in Ag nanoparticle concentration from 0.003 to 0.01 M, respectively.

Increasing the pump pulse energy above the threshold at fixed concentration of nanoparticles, we followed the development of the phase conjugation process characterized by coherent generation of anti-Stokes backscattered signal. Near the threshold, the pulse width of SRMS signal measured by PD is ~3 ns, which is in perfect agreement with the ~150 MHz anti-Stokes spectral shift measured with the use of the Fabry–Perot interferometer. At a 10fold excess of the threshold energy, PD traces of the backward anti-Stokes signal and the pump radiation almost overlaps, which also well agree with the measured spectral shift of ~80 MHz. Dependence of the spectral shift of backward anti-Stokes SRMS signal on the pump pulse energy, determined from the Fabry–Perot interferometry data, is shown in Figure 6. This dependence reveals three stages: In the first stage, the spectral shift is rapidly growing with an increase in pump pulse energy starting from the threshold value. This growth is replaced by the second stage of saturation around the peak value ~150 MHz at moderate pump pulse energies, which is followed by the third stage, the gradual drop to the level of ~80 MHz at a 10fold excess above the threshold in pump pulse energies.

Dependence of the peak backscattered signal intensity on the pump peak intensity is shown in Figure 7. One can find it close to linear from the very threshold, as such the linear dependence is characteristic for the lasing [22], rather than the conventional stimulated temperature scattering effect [8]. In our experiment, we found that the conversion efficiency, i.e., the slope of this dependence, varies from ~0.3 up to 0.7 growing with the concentration of Ag nanoparticles in suspension that varied from 0.003to 0.01 M.

### 3.2. Experimental Results of 2PA SRMS with 100 ps Laser Pulse Width

The scheme of the experiment for picoseconds pulse arrangement (see Figure 4) is at first glance similar to those dealing with nanoseconds, but spectral shifts differ for about two orders of magnitude (see Figure 6 and experimental data below). We performed experiments using Ag nanoparticle suspensions with the same concentrations as in the nanosecond experiment (i.e., 0.003–0.01 M), along with the ZnO nanoparticle suspension (0.001 M concentration). Using this setup, we can compare the results in the sharp focusing (Figure 4a) and in the analog of FWM (Figure 4b) schemes. One can see that near the threshold of SRMS, in the case of mild focusing (Figure 4b), the spectra of reflected light represent a single line anti-Stokes or Stokes shifted, as shown in Figure 8a,c for Ag or ZnO nanoparticle suspensions, respectively. At sharp focusing, the reflected spectrum near the threshold is realized as a Stokes anti-Stokes doublet (see Figure 8b). However, with an increase in pump pulse energy, the doublet transforms into a single line being anti-Stokes shifted for Ag suspension or Stokes shifted for ZnO suspension.

The efficiency of energy transfer from the pump pulse to the backward scattered signal measured in the Ag nanoparticle colloid in the sharp focusing scheme is shown in Figure 9. Starting with the threshold pump pulse energy of ~0.1 mJ and up to ~2 mJ, the 2PA SRMS process reveals rather high efficiencies. With the further increase in pump pulse energies, a strong stimulated Raman scattering in toluene arises which interferes and even completely suppresses the 2PA SRMS signal at pump pulse energies above ~5 mJ.

We found, however, that the Raman process is absent (in our experimental conditions) for the ZnO nanoparticle suspensions in water, which allows us to reach a higher power level of 2PA SRMS scattering signal, as shown in Figure 10. Note that the threshold energy in this case is slightly greater than 1 mJ. We observed the Stokes-shifted scattering dominates for this colloid; however, the anti-Stokes component still remained but did not sufficiently affect the principal results. The efficient reflectivity upon reaching the level 10 mJ was found quite satisfactory.

The dependence of spectral shifts of anti-Stokes backscattered 2PA SRMS signal in Ag nanoparticle colloid (0.003 M) in toluene on the picosecond pump pulse energy is shown in Figure 11. This figure looks similar in shape with the analogous dependence for nanosecond pulse scattering in the same colloid. Note, however, that the shifts values in picosecond scattering are ~50 times larger than those for nanosecond pulses. The maximum value is ~0.25 cm^−1^ which is almost as high as for the SBS frequency shift but has the opposite sign (in Ag nanoparticle colloid in toluene, SBS reveals as the Stokes shifted signal while 2PA SRMS reveals as the anti-Stokes signal). For the points at sufficiently high pump pulse energies in the right-hand side of Figure 11, the signal and the pump spectra almost overlap with each other but can be rather well separated on interferogram due to sufficiently high power of the backscattered signal.

## 4. Discussion

### 4.1. Model of the Nonlinear Medium Response under 2PA SRMS

In this section, we will discuss the above experimental results on 2PA SRMS in nanocolloids in view of the general approach to the theoretical analysis of stimulated temperature scattering, based on the coupled density and temperature fluctuation model. This approach has been developed by Herman and Gray in 1967 for the case of linearly absorbing materials [23,24] and recently modified for two-photon absorption scattering media [12]. According to this model, the density and the temperature fluctuations in nanoparticle colloids irradiated by the laser field of the total strength E are guided by the following equations
(3)$$\begin{array}{l} \frac{\partial^{2}}{\partial t^{2}}\delta \rho -\frac{c_{0}^{2}}{\gamma }\nabla^{2}\delta \rho -\frac{\eta }{\rho_{0}}\frac{\partial }{\partial t}\nabla^{2}\delta \rho -\frac{c_{0}^{2}\rho_{0}\beta_{T}}{\gamma }\nabla^{2}\delta T=-\frac{\gamma^{e}}{8\pi }\nabla^{2}E^{2}\\ \rho_{0}C_{v}\frac{\partial }{\partial t}\delta T-\lambda_{T}\nabla^{2}\delta T-\frac{C_{v}(\gamma -1)}{\beta_{T}}\frac{\partial }{\partial t}\delta \rho ={\left(\frac{n_{0}c}{4\pi }\right)}^{2}\beta E^{4} \end{array},$$
where $\rho_{0}$ is the unperturbed density of the medium, $n_{0}$ is the refractive index and β is the two-photon absorption coefficient, *c* is the speed of light, $c_{0}$ is the velocity of sound, $\lambda_{T}$ and $\beta_{T}$ are the coefficients of heat conduction and thermal expansion at constant pressure, η is the bulk viscosity coefficient, $\gamma =C_{p}/C_{v}$ is the ratio of the specific heats at constant pressure and volume, $\gamma^{e}=\rho_{0}{\left(\partial \varepsilon /\partial \rho \right)}_{T}$ is the electrostriction coefficient, ε is the dielectric susceptibility. The right-hand side term in the first line of Equation (3) describes the electrostriction effect which drives the Brillouin scattering processes, while the temperature oscillations are caused by the two-photon absorption effect which is represented by the right-hand side term in the second line of Equation (3) and drives the 2PA SRMS process of our particular interest.

Density and temperature oscillations are self-consistently coupled to the pump and scattered light fields through the nonlinear polarization of the scattering medium
(4)$$P_{NL}=\frac{1}{4\pi }\left({\left(\partial \varepsilon /\partial \rho \right)}_{T}\delta \rho +{\left(\partial \varepsilon /\partial T\right)}_{\rho }\delta T\right)E,$$
and evolution of the pump and signal waves is then governed by the wave equation
(5)$$n_{0}^{2}\frac{\partial^{2}E}{\partial t^{2}}-\Delta E=\frac{4\pi }{c^{2}}\frac{\partial^{2}P_{NL}}{\partial t^{2}}.$$

The dielectric susceptibility of nanoparticle suspension is described through the Clausius–Mossotti formula
(6)$$\frac{\varepsilon -1}{\varepsilon +2}=\frac{4\pi }{3}\sum_{}N_{j}\alpha_{j},$$
where *N_j_* and *α*_j_ are the number density and the polarizability of molecules and nanoparticles of all types *j* contained in a colloidal solution.

All the above experiments with both the nanosecond and picosecond laser pulses are based on the four-wave mixing schemes. To get visual analytic results, we will restrict ourselves to the limit of collinear wave vectors of all the pump and scattered waves which, in fact, reduces the problem to 1D geometry: all the waves are plane and propagate along the *z* axis (see Figure 2). Additionally, throughout all the analysis below, the small signal approximation is assumed along with negligible depletion of pump radiation.

Both in nanosecond and picosecond interaction regimes, the measured frequency shifts are well below the characteristic Brillouin frequency $\Omega_{B}={\left(c_{0}^{2}q^{2}/\gamma \right)}^{1/2}$, and, assuming the electrostriction effect is small, we find from the first line of Equation (3) that the density perturbation under our experimental condition adiabatically follows the temperature perturbation,
(7)$$\delta \rho \approx \rho_{0}\beta_{T}\delta T,$$
and the second line of Equation (3) acquires the following form
(8)$$\rho_{0}C_{v}(2-\gamma )\frac{\partial }{\partial t}\delta T-\lambda_{T}\nabla^{2}\delta T={\left(\frac{n_{0}c}{4\pi }\right)}^{2}\beta E^{4}.$$

### 4.2. Theory of 2PA SRMS through Four-Wave Mixing in Nanosecond Regime

In the nanosecond domain of pulse widths, the measured in experiment frequency shifts of anti-Stokes and Stokes signals are sufficiently large, $\Omega \tau_{p}>>1$, so they are well separated in the frequency domain from the pump laser pulses. Keeping in mind the experimental four-wave mixing scheme (see Figure 2), the total electric field strength in the interaction region $E=E_{p}^{+}+E_{p}^{-}+E_{s}^{+}+E_{s}^{-}$ is the superposition of two counter-propagating pump waves
(9)$$\begin{array}{l} E_{p}^{+}(z,t)=(1/2)A_{0}\exp[i(\omega_{0}t-k_{0}z)]+c.c.\\ E_{p}^{-}(z,t)=(1/2)a_{0}\exp[i(\omega_{0}t+k_{0}z)]+c.c \end{array},$$
of the same frequency $\omega_{0}$ and wavenumber $k_{0}=n_{0}\omega_{0}/c$, $A_{0}$ and $a_{0}$ are the amplitudes of the incident and backscattered from the rear side of experimental cell pump laser pulses, and two counter-propagating scattered signal waves
(10)$$\begin{array}{l} E_{s}^{+}(z,t)=(1/2)a_{+}(z,t)\exp[i(\omega_{+}t+k_{+}z)]+c.c.\\ E_{s}^{-}(z,t)=(1/2)a_{-}(z,t)\exp[i(\omega_{-}t-k_{-}z)]+c.c \end{array},$$

The scattered anti-Stokes backward propagating $E_{s}^{+}$ and Stokes forward $E_{s}^{-}$ signal are characterized by the amplitudes $a_{\pm }(z)$, the frequencies $\omega_{\pm }$, and the wavenumbers $k_{\pm }\approx n_{0}\omega_{\pm }/c$, respectively. The signal amplitudes are assumed to be slow varying functions of *z* guided by correspondent wave equations which are to be solved self-consistently. We intend to describe the experimentally observed lasing in the four-wave mixing scheme, so that we are seeking conditions for both the coherent amplification and the negative feedback (i.e., self-induced optical cavity) formation inside the interaction region. In doing that, we exclude amplification of scattered signals from noise by imposing the following boundary conditions
(11)$$a_{+}(z=0)=0,a_{-}(z=-L)=0$$
at the ends of this assumed effective cavity 0>z>−L [25].

It is convenient to rewrite using (9) and (10), the total electric field strength as
(12)$$E(z,t)=\frac{1}{2}\left\{\left(A_{0}+a_{+}e^{i\Delta_{+}}\right)\exp[i(\omega_{0}t-k_{0}z)]+(a_{0}+a_{-}e^{-i\Delta_{-}})\exp[i(\omega_{0}t-k_{0}z)]\right\}+c.c.,$$
where the slow phase difference functions are introduced,
(13)$$\begin{array}{l} \Delta_{+}(z,t)=(\omega_{+}-\omega_{0})t+(k_{+}+k_{0})z,\\ \Delta_{-}(z,t)=(\omega_{0}-\omega_{-})t+(k_{0}+k_{-})z. \end{array}$$

After averaging over the optical oscillation period $2\pi /\omega_{0}$, we find
(14)$$\begin{array}{l} \int_{-\pi /\omega_{0}}^{\pi /\omega_{0}}E^{4}(z,t)dt=\frac{3}{8}{\left({\left|A_{0}+a_{+}e^{i\Delta_{+}}\right|}^{2}+{\left|a_{0}+a_{-}e^{-i\Delta_{-}}\right|}^{2}\right)}^{2}+\frac{3}{4}{\left|A_{0}+a_{+}e^{i\Delta_{+}}\right|}^{2}{\left|a_{0}+a_{-}e^{-i\Delta_{-}}\right|}^{2}+\\ +\frac{3}{4}\left({\left|A_{0}+a_{+}e^{i\Delta_{+}}\right|}^{2}+{\left|a_{0}+a_{-}e^{-i\Delta_{-}}\right|}^{2}\right)\left(A_{0}+a_{+}e^{i\Delta_{+}}\right)\left(a_{0}+a_{-}e^{-i\Delta_{-}}\right)e^{-2ik_{0}z}+\\ +\frac{3}{8}{\left(A_{0}+a_{+}e^{i\Delta_{+}}\right)}^{2}{\left(a_{0}+a_{-}e^{-i\Delta_{-}}\right)}^{2}e^{-4ik_{0}z}+c.c. \end{array}.$$

Non-oscillating terms in (14) result in shift in the mean temperature $T_{0}$, so we arrive at the following quasi-stationary temperature oscillation(15)$$\begin{array}{cc} \; & \delta T={\left(\frac{n_{0}c}{4\pi }\right)}^{2}\frac{\beta }{\left(2-\gamma \right)\rho_{0}C_{v}\lambda_{T}q^{2}}\left\{\frac{3}{4}\left({\left|A_{0}\right|}^{2}+{\left|a_{+}\right|}^{2}+2{\left|a_{0}\right|}^{2}+2{\left|a_{-}\right|}^{2}\right)\frac{A_{0}^{*}a_{+}}{1+i\xi }e^{i\Delta_{+}}\right.\\ \; & +\frac{3}{4}\left(2{\left|A_{0}\right|}^{2}+2{\left|a_{+}\right|}^{2}+{\left|a_{0}\right|}^{2}+{\left|a_{-}\right|}^{2}\right)\frac{a_{0}a_{-}^{*}}{1+i\xi }e^{i\Delta_{-}}\\ \; & +\frac{3}{4}\left({\left|A_{0}\right|}^{2}+{\left|a_{+}\right|}^{2}+2{\left|a_{0}\right|}^{2}+{\left|a_{-}\right|}^{2}\right)\frac{A_{0}a_{-}^{*}}{0+i\xi }e^{i(\Delta_{-}-2k_{0}z)}\\ \; & +\frac{3}{4}\left(2{\left|A_{0}\right|}^{2}+{\left|a_{+}\right|}^{2}+{\left|a_{0}\right|}^{2}+{\left|a_{-}\right|}^{2}\right)\frac{a_{0}^{*}a_{+}}{0+i\xi }e^{i(\Delta_{+}-2k_{0}z)}\\ \; & +\frac{3}{4}\left(2{\left|A_{0}\right|}^{2}+{\left|a_{+}\right|}^{2}+2{\left|a_{0}\right|}^{2}+{\left|a_{-}\right|}^{2}\right)\frac{a_{+}a_{-}^{*}}{2+i\xi }e^{i(\Delta_{+}+\Delta_{-}-2k_{0}z)}\\ \; & \left.+\frac{3}{4}\left({\left|A_{0}\right|}^{2}+2{\left|a_{+}\right|}^{2}+{\left|a_{0}\right|}^{2}+2{\left|a_{-}\right|}^{2}\right)A_{0}^{*}a_{0}e^{2ik_{0}z}\right\}+c.c. \end{array},$$

In Equation (15), asterisk (*) means complex conjugation, $\Omega_{T}=4\lambda_{T}k_{0}^{2}/(2-\gamma )\rho_{0}C_{v}$ is the characteristic frequency shift of the temperature scattering, $\xi =\Omega /\Omega_{T}$ is the normalized frequency shift, $\Omega =\omega_{+}-\omega_{0}=\omega_{0}-\omega_{-}$ (frequency shifts of forward and backward signal waves equal each other in the resonance).

One can easily see that only the first four terms of temperature oscillations (15) result in resonance terms of nonlinear polarization *P_NL_*. These terms represent two types of resonance temperature fluctuations: the first kind (the first and the second terms in (15)) corresponds to the slow temperature wave ~exp[±i(Ωt+qz)] with the beat frequency $\Omega =\omega_{+}-\omega_{0}=\omega_{0}-\omega_{-}$ and the wavenumber $q=2k_{0}\approx k_{+}+k_{0}\approx k_{0}+k_{-}$ propagating in the negative direction of the axis *z* along with the anti-Stokes signal. This is a dynamical thermal space grid emerging as a result of the interference between the counter-propagating pump and signal waves. It provides efficient self-consistent distributed feedback through the Bragg reflection of amplified waves [26]. The second kind of temperature perturbation (the third and the fourth terms in Equation (15)) ~exp[±i(Ωt+Δkz)] oscillates also at the beat frequency Ω but the wavenumber is $\Delta k=|k_{\pm }-k_{0}|$. Note that at our experimental conditions with the measured frequency shifts of ~100 MHz, one can estimate ΔkL<<1. In fact, these terms describe homogeneous coherent temperature oscillations which provide coherent amplification, i.e., coherent energy transfer, from pump to signal waves in the four-wave mixing process.

Within the slowly varying amplitude and small signal approximations, keeping only the first order in signal amplitudes terms, we arrive at the following coupled equations for the backward and forward signals
(16)$$\begin{array}{l} \frac{\partial a_{+}}{\partial x}=\left(\frac{1+2\chi }{\xi -i}+\chi \frac{2+\chi }{\xi }\right)a_{+}+\left(\frac{2+\chi }{\xi -i}+\frac{1+2\chi }{\xi }\right)\frac{A_{0}a_{0}}{|A_{0}{|}^{2}}a_{-}^{*}\\ \frac{\partial a_{-}^{*}}{\partial x}=\left(\frac{1+2\chi }{\xi }+\chi \frac{2+\chi }{\xi -i}\right)a_{+}+\left(\frac{2+\chi }{\xi }+\frac{1+2\chi }{\xi -i}\right)\frac{A_{0}^{*}a_{0}^{*}}{|A_{0}{|}^{2}}a_{+} \end{array},$$
where $G=(3/16)\left({\left(\partial \varepsilon /\partial T\right)}_{\rho }-\rho_{0}\beta_{T}{\left(\partial \varepsilon /\partial \rho \right)}_{T}\right)(\beta I_{0}^{2}/k_{0}\lambda_{T})$ is the characteristic gain coefficient, x=Gz is the normalized coordinate, $I_{0}=(c/8\pi )A_{0}^{2}$ is the pump laser intensity, $\chi =|a_{0}{|}^{2}/|A_{0}{|}^{2}$ is the ratio of backward and forward pump intensities. In our derivation of (12), it was assumed $\Omega <<\omega_{0}$ and ΔkL<<1.

We seek solutions in the form $a_{\pm }\sim \exp(-i\lambda x)$ which result in the following characteristic equation of the system (16)
(17)$$\lambda^{2}+(1+4\chi +4\chi^{2})\left(\frac{1}{\xi }+\frac{1}{\xi -i}\right)\lambda +\frac{\left(1-\chi )(1-\chi^{3}\right)}{\xi (\xi -i)}=0$$

In the domain of small pump intensity ratio χ<<1, we find two eigen values which are situated near the points $\lambda_{1}^{0}=i/\xi$ and $\lambda_{2}^{0}=1/\left(\xi -i\right)$. To the first order in *χ* we find
(18)$$\lambda_{1}^{1}\approx \frac{i}{\xi }+\frac{4+9i\xi }{\xi }\chi ,\lambda_{2}^{1}\approx \frac{i}{\xi -i}\left(1+\frac{9\xi -5i}{\xi }\chi \right),$$

To satisfy the boundary conditions (11), solutions to Equation (16) are written as follows
(19)$$\begin{array}{l} a_{+}(z)=C\left(\exp[-iG\lambda_{1}z]-\exp[-iG\lambda_{2}z]\right),\\ a_{-}^{*}(z)=D\left(\exp[-iG\lambda_{1}(z+L)]-\exp[-iG\lambda_{2}(z+L]\right). \end{array}$$

Substituting (19) into (16) we arrive at the relations
(20)$$\begin{array}{l} \frac{D}{C}=\left(\frac{1+2\chi }{\xi -i}+\chi \frac{2+\chi }{\xi }+i\lambda_{1}G\right){\left(\left(\frac{2+\chi }{\xi -i}+\frac{1+2\chi }{\xi }\right)\frac{A_{0}a_{0}}{|A_{0}{|}^{2}}\right)}^{-1}\exp[i\lambda_{1}GL],\\ \exp[i(\lambda_{2}-\lambda_{1})GL]=\frac{\left(\frac{1+2\chi }{\xi -i}+\chi \frac{2+\chi }{\xi }+i\lambda_{1}G\right)}{\left(\frac{1+2\chi }{\xi -i}+\chi \frac{2+\chi }{\xi }+i\lambda_{2}G\right)}. \end{array}.$$

The first of these relations gives the ratio of Stokes forward and anti-Stokes backward signal amplitudes, while the second equation results in the lasing conditions. After substitution of the eigen values (18) and separation of real and imaginary parts, we find the following relations
(21)$$\begin{array}{l} GL=(1+\xi^{2})\tan^{-1}\left(-\frac{9\xi }{\xi^{2}+1}\right),\\ \chi^{-1}=\sqrt{{\left(9\xi^{2}-2\right)}^{2}+9\xi^{2}}\exp\left[\frac{1}{\xi }\tan^{-1}\left(-\frac{9\xi }{\xi^{2}+1}\right)\right], \end{array}$$
which must be satisfied simultaneously to achieve self-consistent cavity formation and the lasing threshold. The second of these equations relates the normalized frequency shift *ξ* and the ratio of the intensities of pump backward and forward waves. The first equation allows one to determine the SRMS threshold: the effective cavity length cannot be larger than the experimental cell length, so one should provide a sufficiently large gain coefficient (which is proportional to the squared pump intensity) to fit these relations.

Considering the lasing conditions (21), one can easily find that zero frequency shift is strictly prohibited. Moreover, 2PA SRMS inevitably exhibits an inherent nonzero frequency shift, and its minimum value is $\xi_{\min}=\sqrt{2}/3$, i.e., $\Omega_{\min}\approx 0.471\lambda_{T}q^{2}/(2-\gamma )\rho_{0}C_{v}$.

### 4.3. Theory of Picosecond 2PA SRMS Backward Signal Amplification

In contrast to the quasi-continuous character of 2PA SRMS processes for nanosecond laser pulses, in the picosecond domain of pulse widths we have $\Omega \tau_{p}>1$, so that the pump and the signal spectra considerably overlap each other and it is not possible to separate spectrally the forward pump wave and the forward scattered signal from each other, as well as backward signal and backward pump waves. In this case, instead of Equation (12), it is convenient to make the following replacement
(22)$$\begin{array}{l} A_{0}+a_{-}\exp(-i[\Omega t+\delta kz])=A_{p}(z,t)\\ a_{0}+a_{+}\exp(i[\Omega t+\delta kz])=A_{s}(z,t) \end{array},$$
where again $\Omega =\omega_{+}-\omega_{0}=\omega_{0}-\omega_{-}$ and $\delta k=k_{+}-k_{0}\approx k_{0}-k_{-}$ are the frequency shift and wavenumber difference, and rewrite the total field strength as
(23)$$E(z,t)=\frac{1}{2}\left\{A_{p}(z,t)\exp[i(\omega_{0}t-k_{0}z)]+A_{s}(z,t)\exp[i(\omega_{0}t-k_{0}z)]\right\}+c.c.$$

The experiment spectral shifts measured in picoseconds are still lower than the characteristic Brillouin frequency but considerably larger than the characteristic thermal frequency shift
(24)$$4\lambda_{T}k_{0}^{2}/\rho_{0}C_{v}=\Omega_{T}<<\Omega <<\Omega_{B}={\left(c_{0}^{2}q^{2}/\gamma \right)}^{1/2},$$

Then, the temperature evolution equation now appears as follows
(25)$$\rho_{0}(2-\gamma ){С}_{v}\frac{\partial T}{\partial t}\approx \beta {\left(\frac{n_{0}c}{4\pi }\right)}^{2}E^{4}.$$

In view of Equation (14), the slow oscillating terms in the right-hand side of Equation (25) are
(26)$$\int_{-\pi /\omega_{0}}^{\pi /\omega_{0}}E^{4}(z,t)dt=\frac{3}{4}(|A_{p}{|}^{2}+|A_{s}{|}^{2})\left(A_{p}^{*}A_{s}\exp(iqz)+A_{p}A_{s}^{*}\exp(-iqz)\right).$$

We again assume signal amplitude is small compared to the pump amplitude and find in the first order in perturbation theory the temperature oscillates as
(27)$$T\approx \frac{3}{4}\frac{\beta }{(2-\gamma )\rho_{0}C_{v}}{\left(\frac{n_{0}c}{4\pi }\right)}^{2}\exp(iqz)\int_{-\infty }^{t}dt\prime |A_{p}{|}^{2}A_{p}^{*}A_{s}+c.c.$$

Keeping only the resonance terms in the nonlinear polarization, we find from the wave Equation (5) under the slow varying amplitude approximation the following transient equation for the signal amplitude
(28)$$\frac{\partial A_{s}}{\partial z}-\frac{1}{v_{0}}\frac{\partial A_{s}}{\partial t}=-iBA_{p}\int_{-\infty }^{t}dt\prime |A_{p}{|}^{2}A_{p}^{*}A_{s},$$
with the coefficient *B* given by
(29)$$B=\frac{3k_{0}c^{2}}{8{\left(4\pi \right)}^{2}}\frac{\beta \beta_{T}}{(2-\gamma )C_{v}}\left[{\left(\frac{\partial \varepsilon }{\partial \rho }\right)}_{T}-\frac{1}{\rho_{0}\beta_{T}}{\left(\frac{\partial \varepsilon }{\partial T}\right)}_{\rho }\right].$$

To obtain qualitative analytical consideration, we will assume below that the pump pulse amplitude remains approximately constant $A_{p}(z,t)=A_{0}\approx const$, i.e., its temporal shape is rectangular with the pulse width $\tau_{p}$, and both the pump and the signal pulses are characterized by equal group velocities $v_{0}$. Making conventional coordinate transformation to the moving frame $\left(z,t)\to (z,\zeta =t-z/v_{0}\right)$, *ζ* is the intrinsic eigen time coordinate of the signal pulse. Let us split the signal amplitude into two parts, $A_{s}(z,\zeta )=a_{0}(\zeta )+a_{s}(z,\zeta )$, where $a_{s}(z,\zeta )$ is the nonlinear part of the signal amplitude, which emerges as a result of stimulated backscattering, and $a_{0}(\zeta )=a_{0}f(\zeta )$ is the amplitude of backward linearly reflected part of the pump pulse, $a_{0}=\kappa A_{0}$, κ is the linear reflection coefficient, $|\kappa {|}^{2}=\chi$. The temporal rectangular shape function is f(ζ)=1 at $0<\zeta <\tau_{p}$ and f(ζ)=0 outside this interval, at ζ<0 and $\zeta >\tau_{p}$. Introducing the envelope pulse area function
(30)$$\Phi_{s}(z,\zeta )=\int_{0}^{\zeta }a_{s}(z,\zeta )d\zeta ,$$
we arrive at the following equation
(31)$$\frac{\partial^{2}\Phi_{s}}{\partial z\partial \eta }+K\Phi_{s}=-Ka_{0}f(\eta )(\eta -z/v_{0})g(z)\;,$$
where the coefficient $K=iB|A_{0}{|}^{4}$, and the function *g*(*z*) accounts for the finite length of the interaction region, *g*(*z*) = 1 at −L<z<0 and vanishes outside this region. Natural initial and boundary conditions are $\Phi_{s}(\zeta ){|}_{z=0}=0$ and $\Phi_{s}(z){|}_{\zeta =0}=0$, i.e., no seed of nonlinear signal is assumed at the entrance *z* = 0 of the interaction region (note, the signal propagates counter to the pump pulse entering the interaction region at *z* = 0 and leaving it at *z* =−*L*).

For our experimental conditions, the interaction region length is small with respect to the pulse width, $L<<c\tau_{p}$, so that one can replace $\zeta -z/v_{0}\to \zeta$ in the right-hand side of Equation (31). The Green function of this equation is then given by the Riemann function [27] and we find
(32)$$\Phi_{s}(z,\eta )=-Ka_{0}\int_{0}^{\zeta }ydy\int_{0}^{z}J_{0}(\sqrt{4K(z-\xi )(\zeta -y)})d\xi ,$$
where *J*_0_ (x) is the Bessel function of zero order. Finally, calculating the nonlinear signal amplitude, we find
(33)$$a_{s}(z,\eta )=\frac{{\partial \Phi }_{s}}{\partial \eta }=a_{0}\left(J_{0}(\sqrt{4Kz\zeta })dt-1\right).$$

One can easily see that at sufficiently low pump intensities $B|A_{0}{|}^{4}L\tau_{p}<<1$ the scattered signal exhibits linear growth along the interaction region,
(34)$$a_{s}\approx -Ka_{0}z\zeta ,$$

With an increase in pump intensity, a highly nonlinear interaction regime emerges when the argument of the Bessel function becomes large. Taking into account asymptotic approximation of the Bessel function [28], $J_{0}(z)\sim \sqrt{2/\pi z}\cos(z-\pi /4)$, we arrive at the following formula for the signal amplitude
(35)$$a_{s}(z,\zeta )\sim \frac{a_{0}}{i^{1/4}{\left(16\pi^{2}B|A_{0}{|}^{4}L\zeta \right)}^{1/4}}\exp\left((1+i)\sqrt{2B|A_{0}{|}^{4}L\zeta }\right)$$

It demonstrates exponent-like growth with the rate reaching their maximum by the pulse tail $\zeta \to \tau_{p}$,
(36)$${\left|a_{s}/a_{0}\right|}^{2}\approx \frac{\exp[\Gamma ]}{\sqrt{2}\pi \Gamma },$$
where the characteristic nonlinear gain coefficient is $\Gamma =2\sqrt{2BL\tau_{p}}|A_{0}{|}^{2}$, and the characteristic shift in spectrum can be determined as
(37)$$\Omega \sim {\left(B|A_{0}{|}^{4}L/2\tau_{p}\right)}^{1/2},$$

Thus, our theory recovers linear growth of the spectral shift with the pump intensity near the threshold which we observe in the picosecond experiment (see Figure 11). With an increase in pump intensity, the reflection coefficient (signal-to-pump ratio of the peak amplitudes) is also rapidly growing (see Figure 8 and Figure 9) so that the first order perturbation theory becomes insufficient to describe the saturation of the frequency shift as well as the following drop, possibly due to significant pump pulse depletion which is outside of our model.

## 5. Conclusions

In conclusion, we reported here the results of our experimental and theoretical study of 2PA SRMS in nanosecond-to-picosecond domain of laser pulse widths. Our measurements were taken in suspensions of metal~(Ag) nanoparticles in toluene (both in nanosecond and picosecond domain) and ZnO nanoparticles in distilled water (at picosecond pulse width). With the use of 12 ns 0.527 μm laser pulses, we demonstrated the 2PA SRMS threshold of ~0.2 mJ and the non-zero frequency upshift of the backscattered anti-Stokes signal in the order of 100 MHz in the Ag nanoparticle suspension in toluene. The frequency shift rapidly grows with an increase in pump pulse energy, reaches saturation at ~150 MHz, and then decreases to ~80 MHz at 10fold threshold pump pulse energy. We treat this process as coherent amplification of scattered signals resulting from the four-wave mixing between two (forward and backward) pump waves and two counter-propagating Stokes and anti-Stokes signal waves. Coupling between these four waves emerges through the nonlinear polarization caused by two-photon excited temperature oscillations (along with correspondent density oscillations). We show that there are two kinds of these temperature oscillations, namely spatially homogeneous oscillations at the frequency equal to the anti-Stokes spectral shift, and the second is the slow temperature wave resulting from the interference between pump and signal waves. Amplification in this system is caused mainly by coherent temperature oscillations, while the slow temperature wave provides a dynamic spatial grid which results in a self-organized distributed-feedback optical cavity. We derived the lasing condition in this FWM scheme and found the gain and the length of the cavity and the frequency shift of the Stokes and anti-Stokes components near the threshold. The developed theoretical approach allows us to show that lasing can be achieved only for non-zero frequency shifts exceeding a specific minimum value.

For the first time, we reported that the efficient 2PA SRMS is achieved in Ag nanoparticle suspensions in toluene and in ZnO nanocolloids in water for 100 ps 0.532 μm laser pulses. Scattering threshold pulse energy in Ag nanocolloids in toluene is measured as ~0.1 mJ, while the threshold for the ZnO nanocolloid in water is measured larger by an order of magnitude. We observe that the anti-Stokes spectral shift in Ag nanocolloid behaves with an increase in pump pulse energy in the same manner as for nanosecond pulses, but shifts are ~50 times larger reaching values up to 0.25 cm^−1^ which are close to the SBS spectral shifts. The scattering efficiency (signal-to-pump ratio of pulse energies) is measured to be very high reaching ~50% in Ag and even above in ZnO nanocolloids, which make the 2PA SRMS very prospective for phase conjugation in the picosecond domain. We extended our four-wave mixing model of 2PA SRMS into the picosecond time domain. In contrast with the nanosecond domain, where the pump and the forward and backward scattered signals are well separated in frequencies, the spectra of pump and signal waves significantly overlap at picosecond 2PA SRMS. As a result, only the dynamic temperature grating provides the scattering and amplification effects making the picosecond 2PA SRMS more akin the parametric amplification process.

Concluding the above our findings, 2PA SRMS in nanocolloids was revealed as a prospective candidate for moving the efficient PC technique into the range of shorter laser pulses and decreasing energy (power) thresholds.

## Figures and Tables

**Figure 1 nanomaterials-12-02567-f001:**
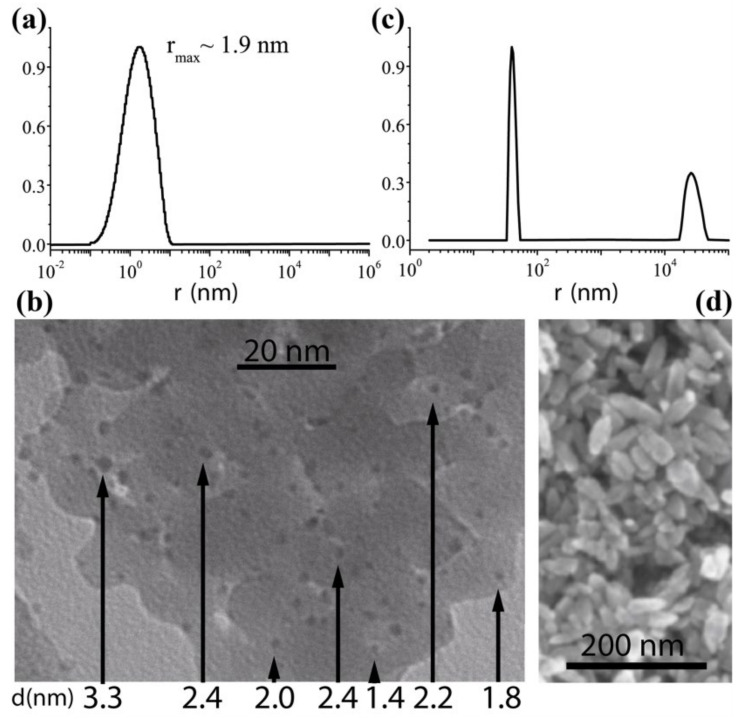
The results of Ag (**a**,**b**) and ZnO (**c**,**d**) nanoparticles characterization: (**a**,**c**) characteristic size measurements using the correlation spectroscopy method processed with the use of DynaLS software; (**b**,**d**) scanning transmission electron microscope patterns along with the diameter values extracted for several nanoparticles.

**Figure 2 nanomaterials-12-02567-f002:**
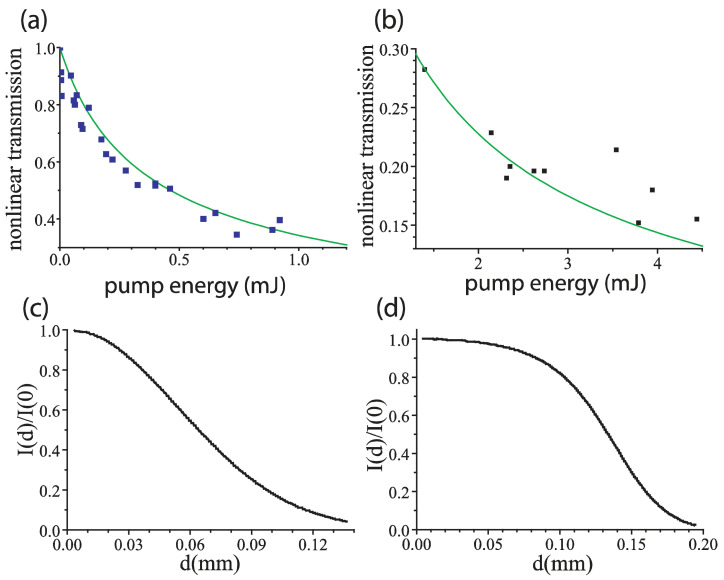
Nonlinear transmission of the 100 ps 0.532 μm laser radiation in the 5 cm-thick layer of Ag nanoparticle suspension measured in two domains of pulse energies, (**a**) below 1 mJ, (**b**) 1÷4.5 mJ. Experimental points are given by rectangles. Normalized output radial intensity profiles are shown below the plots: (**c**) low-distorted near-Gaussian profile at low pulse energies and (**d**) heavily distorted profile at 1÷4.5 mJ pulse energies. Green lines in (**a**,**b**) show the fitting according to Equation (2).

**Figure 3 nanomaterials-12-02567-f003:**
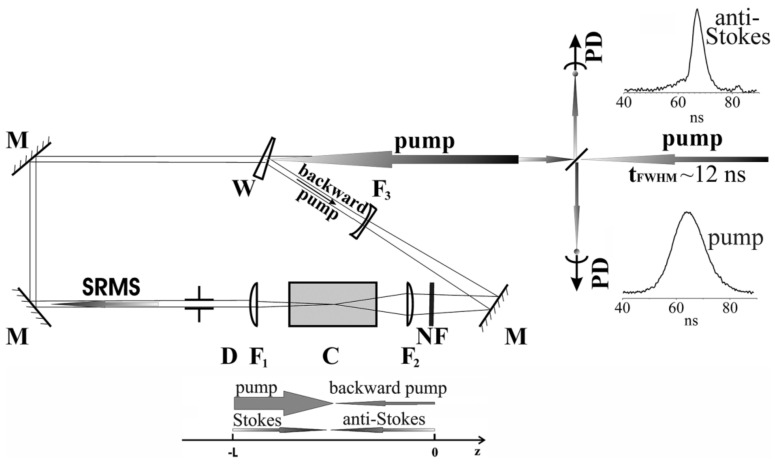
Experimental setup for nanosecond four-wave mixing SRMS generation. It contains: 5 cm interaction cell (C), lenses with the focal lengths 7.5, 5.5, and −15 cm (F_1−3_, respectively), total reflection mirrors (M), the glass wedge (W), the 4 mm aperture (D), the neutral density filter (NF), and two photodiodes PD of ~1 ns resolution. Typical oscilloscope traces of the pump and Anti-Stokes signal pulses are shown at correspondent PD in the right upper corner. The direction map of the pump pulses and scattering signals is shown below the cell C.

**Figure 4 nanomaterials-12-02567-f004:**
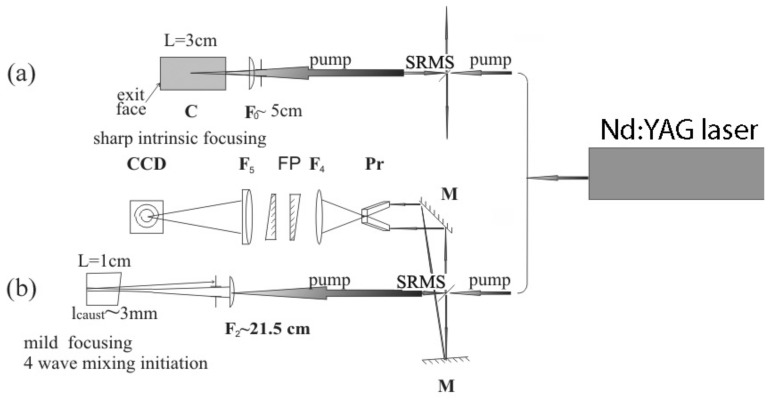
Experimental setup for 2PA SRMS of 100 ps pulses. There are two interaction geometry options, (**a**) sharp focusing of pump radiation into the interaction cell of 3 cm length with the lens of 5 cm focal length, and (**b**) mild focusing into short 1 cm interaction cell with the lens of 21.5 cm focal length, providing the four-wave mixing process due to the backward reflection of the pump pulse by the rear side of the cell. The upper part in (**b**) shows the scheme of Fabry–Perot spectral measurements: Pr—waveguide prism; F_4-5_—objectives transcribing the edge of the prism on the CCD matrix plane.

**Figure 5 nanomaterials-12-02567-f005:**
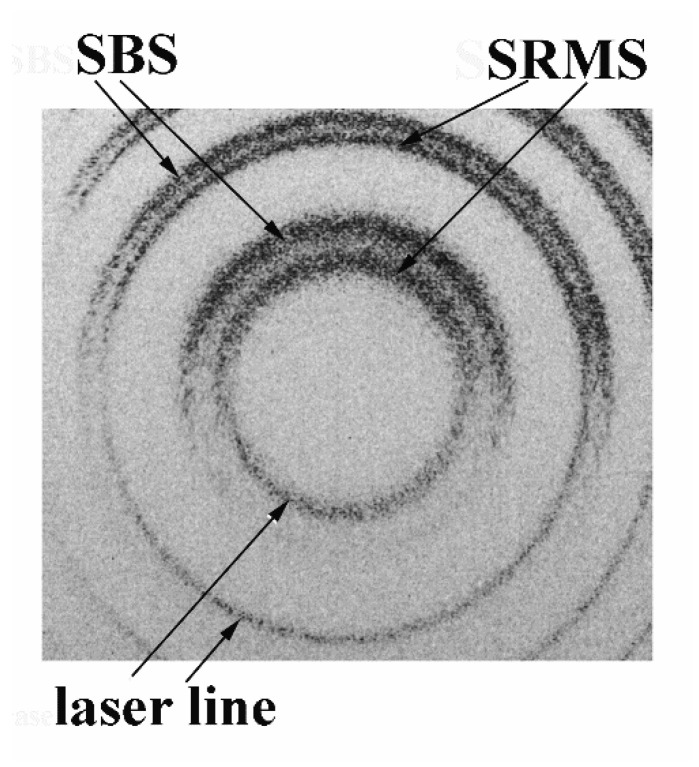
Typical Fabry–Perot spectra in the case of nanosecond pulses: pump (**lower part**), backscattered Brillouin and Rayleigh signals (**upper part**).

**Figure 6 nanomaterials-12-02567-f006:**
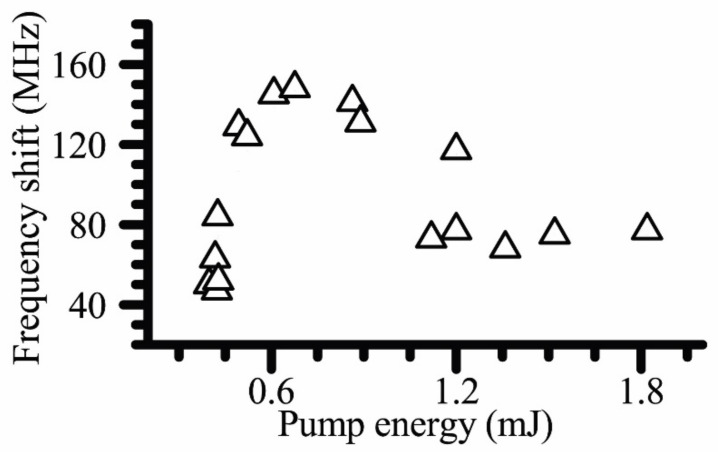
Spectral shifts of the anti-Stokes backward SRMS signal vs the pump pulse energy in the nanosecond pulse interaction regime, experimental points are given by triangles.

**Figure 7 nanomaterials-12-02567-f007:**
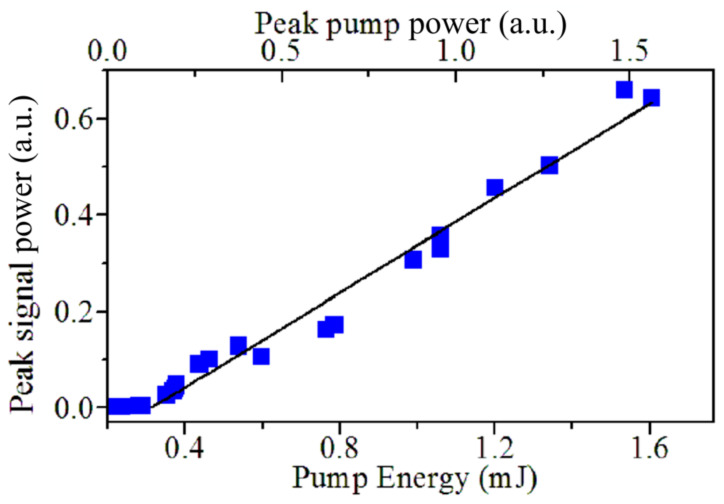
Efficiency of backward SRMS reflection vs. the pump pulse energy in the nanosecond interaction regime, which is measured as the peak-to-peak ratio of PD signals. Experimental points are shown by rectangles. Threshold pump pulse energy is ~0.3 mJ, the slope is ~0.4.

**Figure 8 nanomaterials-12-02567-f008:**
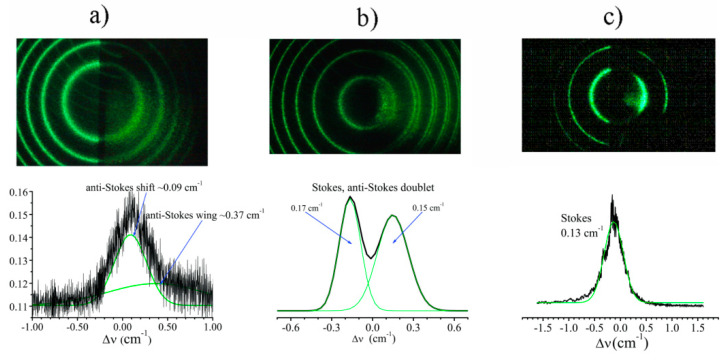
SRMS backscattering light spectra initiated by 100 picoseconds pulses in Ag nanoparticle colloids in toluene (**a**) and ZnO nanoparticle colloids in water (**b**,**c**). The results are given for the following experimental schemes: (**a**) four-wave mixing scheme at the mild focusing case (Figure 4b with F_2_ = 21.5 cm): (**b**) sharp focusing scheme (Figure 4a with F_0_ = 5 cm); (**c**) scheme of Figure 4a with the replaced lens of F_0_ = 15 cm focal length.

**Figure 9 nanomaterials-12-02567-f009:**
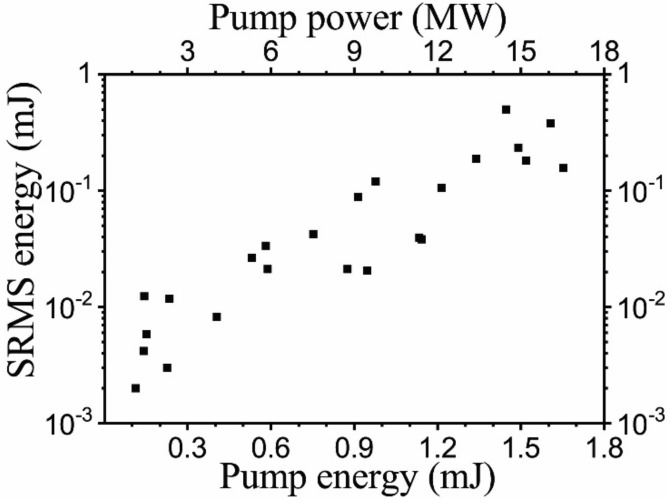
Backscattering efficiency for 0.003 M Ag nanoparticle solution in toluene for the 100 ps pump laser pulse, the threshold is ~0.1 mJ. Experimental points are shown by rectangles.

**Figure 10 nanomaterials-12-02567-f010:**
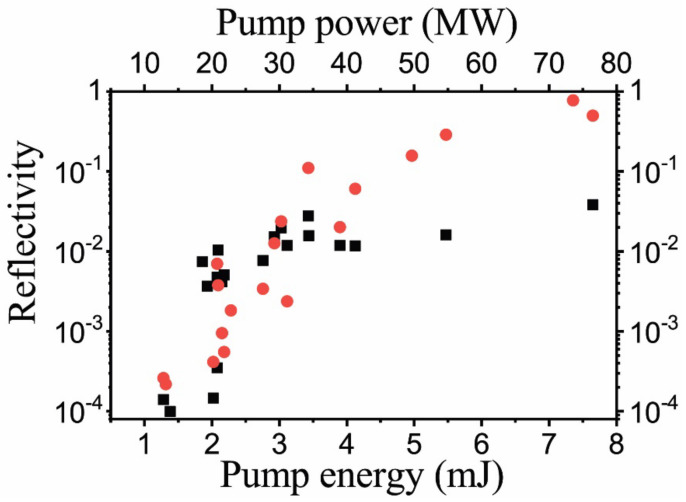
Reflectivity (signal-to-pump ratio of pulse energies) of ZnO nanoparticle solution in water vs. pump pulse energy measured for the Stokes shifted (red circles) and anti-Stokes shifted backscattered radiation (black squares) from 100 ps pump laser pulses.

**Figure 11 nanomaterials-12-02567-f011:**
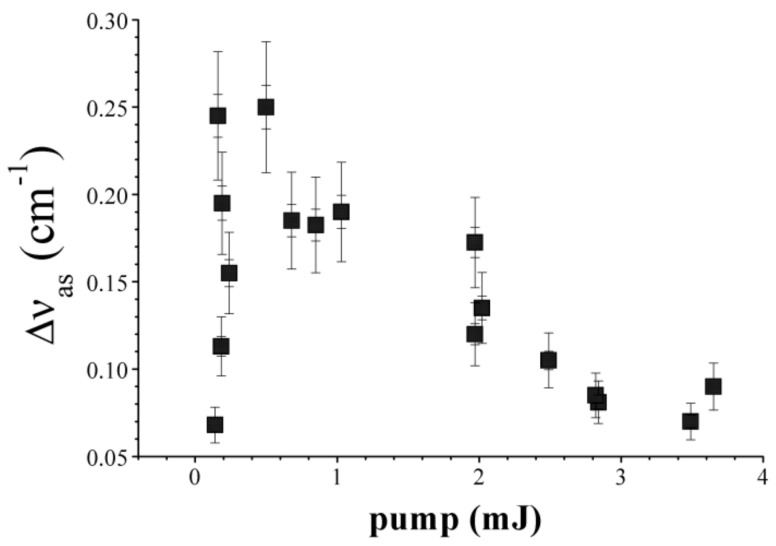
Anti-Stokes 2PA SRMS spectral shifts vs. the pump pulse energy measured in Ag nanoparticle colloid solution (0.003 M) under 100 ps 0.532 μm laser radiation.

## Data Availability

The data presented in this study are available on request from the corresponding author.

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
