# Peer review of "Stimulated Thermal Scattering in Two-Photon Absorbing Nanocolloids under Laser Radiation of Nanosecond-to-Picosecond Pulse Widths"

_nanomaterials, 2022, doi:10.3390/nano12152567_

Round 1

Reviewer 1 Report

The authors investigate the SRMS effects in Ag and ZnO nanocolloids using green-laser-pulse excited FWM, where they claim a stimulated thermal scattering in two-photon absorbing regime. This might be an interesting study and might be suitable for publication. However, it is NOT well written and much improvement is required. The following issues must be addressed before this manuscript can be considered for acceptance.

(1) In the abstract, the authors claim a 532 nm laser was used for all of the investigations, however, they actually used a 527 nm laser for the ns 2PA SRMS. They did not mention the 527 nm excitation in the conclusion either.

(2) I do not agree with the authors for their definition of “2PA” processes simply on the basis of the degenerate FWM using green laser pulses. These processes are simply FWM with two identical green pulses involved as the excitation. They should consider make revisions or give more convincing interpretations and discussions for the “2PA”.

(3) Precise drawings of the photon energy relationships between the excitation and the scattering processes should be included for understanding the FWM processes.

(4) All of the pump properties (Pump pulse energy, Pump pulse width/length, Pump power/peak power, etc.) are displayed as “Pump” in figures 2, 3, 9, 10……, this is unacceptable for the characterization of the excitation pulses.

(5) Control experiments are needed to rule out the possibility that the FWM processes are indeed taking place with the Ag and ZnO nanoparticles, instead of with the medium supporting their suspension and their containing.

(6) What’s the definition of “nonlinear transmission”? This needs to be clarified by a precise formulation.

(7) The experimental data are all poorly presented, which influence the understanding of the physical mechanisms. The format, quality, resolutions of the figures need to be improved substantially.

(8) In Fig. 1(b), there are numbers in different color and different sizes displayed, however, one cannot recognize their correspondence with the particles and locations.

(9) In Fig. 2, for the low panel, the horizontal axis is not defined! Furthermore, there is only one number is displayed, how can one justify the true scaling of the axis?

(10) The solution in (2) is definitely wrong! It can be identified simply by seeing its structure and physical meaning.

(11) Considering the problem in my question (9), I cannot believe completely that the formula and equations in the manuscript are all correct. Please make detailed and full check on all of the formula and equations, also in the theoretical sections, to confirm their correctness.

Reviewer 2 Report

In their paper entitled “Stimulated thermal scattering in two-photon absorbing nanocolloids under laser radiation of nanosecond-to-picosecond pulse widths”, the authors present similar scientific experimental and theoretical concepts already published and detailed in their own previous references [11-12] related to the spectral shift of stimulated Rayleigh-Mie/thermal scattering under two-photon absorption in colloidal solutions of metal nanoparticles. In the present work, the originality lies in the extension to picosecond regime and the simultaneous use of metal (Ag) and ZnO nanoparticles. A detailed experimental and theoretical comparison between the nanosecond and picosecond regime is discussed. The work could reveal interesting to be considered for publication in the Special Issue “Thermophysical Properties of Nanocolloids and Their Potential Applications” of the Journal Nanomaterials but suffers from serious writing flows and mistakes in the references to their Figures in the text to be considered as “readable” at this level.

The authors are encouraged to drastically improve the global English editing of their work as they successfully published in references [11-12] and correct all mistakes in Figures numbering

Among too numerous typo mistakes and not understandable sentences, for instance:

- In section 3.2, there is no coherence between the Figure numbering in the text with the corresponding pictures. Similar issue are encountered in section 4.

- In Figure 10 caption, the blue squares results are not explicitly referenced to the corresponding results.

- Repeated mistakes through the paper in the laser pump wavelength (0.532 nm written instead of 0.532 µm).

- In Figure 1, missing units in the X and Y scale

- Lines 224 and at other places in the text  : “rare wall” makes no sense and should be replaced for instance by “rear side”.

- Lines 280-281 : “Till increase the pump level the doublet transforms in single line be it anti-Stokes shifted for Ag NP or Stokes for ZnO” : no sense sentence

- Figure 8b “dublet”  to be replaced by “doublet”

- The sub-section numbering in section 4 is erroneous

- Line 507 “, while, while”

And so on…

In summary, this paper is at the writing level of a “draft” and should be greatly improved to be considered for publication in Nanomaterials.

Yours Sincerely

Round 2

Reviewer 1 Report

1. If comparing the old and the latest versions of this manuscript, one can find clearly that the equation (2), which is the solution to equation (1), was obviously wrong in the first version, because of the lacking of the important minus symbol "-".

This minus symbol is very critical to the physical meaning of the solution. The authors quietly add the "-" in equation (2) in this revised manuscript, however, they do not admit this and said it was correct. This is what I cannot understand and what I am not satisfied.

Nevertheless, I am happy to see they have corrected the mistake, although they do not want to admit it.

2. Basically, the authors have addressed most of my questions, I recommend acceptance of this manuscript for publication.

Author Response

We deeply apologize to the reviewer for our response, which he regarded as unacceptable. Indeed, in the first version of the article, we omitted the minus sign in Eq. (2). However, this unfortunate typo should not be a reason to doubt all other calculations given in the manuscript.

We express our gratitude to the reviewer that he find our revised manuscript acceptable for publication.

Reviewer 2 Report

The authors have adressed all the issues raised in my previous report.

The paper can now be considered for publication in Nanomaterials after a last careful proof-checking of the text editing and some  minor typo mistakes.

Yours Sincerely

Author Response

We extend deep thanks to the reviewer for approvement of the revised version of our manuscript